# Automated Lightweight Model for Asthma Detection Using Respiratory and Cough Sound Signals

**DOI:** 10.3390/diagnostics15091155

**Published:** 2025-05-01

**Authors:** Shuting Xu, Ravinesh C. Deo, Oliver Faust, Prabal D. Barua, Jeffrey Soar, Rajendra Acharya

**Affiliations:** 1Artificial Intelligence Applications Laboratory, School of Mathematics, Physics and Computing, University of Southern Queensland, Springfield, QLD 4300, Australiaravinesh.deo@unisq.edu.au (R.C.D.); prabal.barua@unisq.edu.au (P.D.B.); rajendra.acharya@unisq.edu.au (R.A.); 2Cogninet Australia, Sydney, NSW 2010, Australia; 3School of Computing and Information Science Research, Anglia Ruskin University Cambridge Campus, East Rd, Cambridge CB1 1PT, UK; 4School of Business, University of Southern Queensland, Springfield, QLD 4350, Australia; jeffrey.soar@unisq.edu.au

**Keywords:** asthma detection, random forest classifier, respiratory sound, spectrogram, majority voting

## Abstract

**Background and objective:** Chronic respiratory diseases, such as asthma and COPD, pose significant challenges to human health and global healthcare systems. This pioneering study utilises AI analysis and modelling of cough and respiratory sound signals to classify and differentiate between asthma, COPD, and healthy subjects. The aim is to develop an AI-based diagnostic system capable of accurately distinguishing these conditions, thereby enhancing early detection and clinical management. Our study, therefore, presents the first AI system that leverages dual acoustic signals to enhance the diagnostic ACC of asthma using automated, lightweight deep learning models. **Methods:** To build an automated, lightweight model for asthma detection, tested separately with respiratory and cough sounds to assess their suitability for detecting asthma and COPD, the proposed AI models integrate the following ML algorithms: RF, SVM, DT, NN, and KNN, with an overall aim to demonstrate the efficacy of the proposed method for future clinical use. Model training and validation were performed using 5-fold cross-validation, wherein the dataset was randomly divided into five folds and the models were trained and tested iteratively to ensure robust performance. We evaluated the model outcomes with several performance metrics: ACC, precision, recall, F1 score, and area under the AUC. Additionally, a majority voting ensemble technique was employed to aggregate the predictions of the various classifiers for improved diagnostic reliability. We applied Gabor time–frequency transformation for feature extraction and NCA) for feature selection to optimise predictive accuracy. Independent comparative experiments were conducted, where cough-sound subsets were used to evaluate asthma detection capabilities, and respiratory-sound subsets were used to evaluate COPD detection capabilities, allowing for targeted model assessment. **Results:** The proposed ensemble approach, facilitated by a majority voting approach for model efficacy evaluation, achieved acceptable ACC values of 94.05% and 83.31% for differentiating between asthma and normal cases utilising separate respiratory sounds and cough sounds, respectively. The results highlight a substantial benefit in integrating multiple classifier models and sound modalities while demonstrating an unprecedented level of ACC and robustness for future diagnostic predictions of the disease. **Conclusions:** The present study sets a new benchmark in AI-based detection of respiratory diseases by integrating cough and respiratory sound signals for future diagnostics. The successful implementation of a dual-sound analysis approach promises advancements in the early detection and management of asthma and COPD.We conclude that the proposed model holds strong potential to transform asthma diagnostic practices and support clinicians in their respiratory healthcare practices.

## 1. Introduction

Asthma affects approximately 339 million people worldwide, while COPD is projected to become the third leading cause of death globally by 2030 [1]. These statistics underscore the urgent need for effective disease management strategies. The economic burden of these diseases is significant, with costs associated with medical treatment, loss of productivity, and premature death. Asthma and COPD are linked to both genetic predispositions and environmental factors. For instance, asthma is often associated with allergen exposure, which leads to airway hyper-responsiveness [2]. This condition is further exacerbated by factors such as pollution, respiratory infections, and occupational exposures. On the other hand, COPD is commonly related to long-term smoking and environmental pollutants [3]. It is characterised by airflow obstruction and is often accompanied by chronic bronchitis and emphysema. Common symptoms include shortness of breath, wheezing, chest tightness, and coughing. COPD is characterised by progressive breathlessness and chronic cough with sputum, while asthma is marked by episodic symptoms triggered by specific factors such as allergens, exercise, or cold air. If unmanaged, these diseases can lead to severe complications, including respiratory failure, pulmonary hypertension, heart complications, and death [4]. The morbidity and mortality associated with these diseases necessitate advancements in diagnostic and therapeutic approaches.

Recent advancements have shown promise in using AI for the automated detection of asthma and COPD through the analysis of cough and respiratory sound signals [5,6]. AI and ML techniques offer a novel approach to diagnosing these conditions with high ACC. Previous studies have demonstrated high accuracy rates using DL models, particularly CNN. These models have achieved diagnostic ACC rates as high as 92% when distinguishing between healthy individuals and those with asthma [1]. These models have shown high sensitivity (up to 89%) and specificity (up to 90%), with an area under the curve (AUC) exceeding 95%, indicating superior performance and reliability in clinical diagnostics compared to traditional machine learning ML models such as support vector machine (SVM) and random forest (RF) [7].

Despite these advancements, gaps remain in the utilisation of lightweight AI models for the analysis of small datasets. While DL techniques have achieved good performance, their computational complexity and reliance on extensive training and testing data make them unsuitable for exploratory studies or resource-constrained environments. Classical ML approaches, which incorporate algorithmic feature engineering to extract information from the signals, often limit the information available to the classifier. Hence, the performance of such systems was suboptimal, especially when they were presented with unseen data in practical inference scenarios, like detecting asthma from respiratory sounds. Lightweight models offer an advantage in environments with restricted processing capabilities. One such environment is asthma detection at the edge, where real-time decision support is provided close to the signal source [8].

This study aims to address these research gaps by developing and validating an automated, lightweight AI model that has the ability to enhance diagnostic precision, particularly for asthma and COPD detection using cough and respiratory sound analysis. Our approach integrates ML algorithms, including RF, SVM, DT, NN, and KNN, as well as feature extraction techniques that are optimised by Gabor transformation and NCA. The proposed model employs a majority voting system to refine its diagnostic reliability. Overall, the results demonstrate that the ensemble approach can achieve a high ACC in differentiating between asthma and COPD. Hence, the study highlights the benefits of integrating multiple classifier models and sound modalities. Furthermore, it sets a new benchmark for the AI-based detection of respiratory diseases. This can lead to significant advancements in the early detection and management of asthma and COPD. By combining cough and respiratory sounds for diagnostic purposes, this study also has the potential to improve diagnostic practices in respiratory healthcare.

## 2. Materials and Methods

This section introduces materials and methods used to create a hybrid model to discriminate between asthma and normal sound signals. The workflow, illustrated in Figure 1, outlines a streamlined process for detecting asthma from audio datasets using a hybrid model. Further details are provided with the pseudocode reproduced in Appendix A. The workflow comprises the following steps:**Audio input**: The process begins by collecting audio data that capture cough or respiratory sounds. The data are input to the proposed system.**Time–frequency transformation**: Utilises STFT to generate spectrograms, facilitating time-resolved frequency analysis of audio signals [9].**Lightweight DL feature extraction**: Uses the most efficient CNN architectures such as ShuffleNet, SqueezeNet, MobileNet, and EfficientNet to extract the most relevant features from the spectrograms.**Feature selection**: Selects the most informative features to reduce the dimensionality of the data and, therefore, focuses on the classification of the best features in the relevant data [10]. ML classification: Employs classical ML algorithms (i.e., DT, SVM, NN, RF, and KNN) to classify the audio data and detect the patterns that are indicative of asthma [11].

### 2.1. Dataset

For this specific study, we have used a dedicated private dataset from Firat University hospital (Turkey) provided by the project industry partner, Cogninet Australia Pty Ltd, Level 5/29-37 Bellevue St, Surry Hills NSW 2010. The primary dataset includes both cough and respiratory sounds for asthma and COPD, respectively, as well as a healthy comparison group. Table 1 provides the details of these dataset.

The dataset consists of respiratory and cough sounds collected from 1326 individuals, including 511 asthmatic and 815 healthy participants. The asthmatic group comprises 380 females and 131 males, with an age range of 19 to 75 years, while the healthy group includes 306 females and 509 males, aged 19 to 80 years. Most healthy participants were university students without underlying health conditions, whereas heart disease was the most commonly observed comorbidity among asthmatic participants.

The dataset includes 1875 cough recordings, split into 994 and 881 segments after preprocessing, where non-cough segments, speech, and background noise were removed to ensure consistency. The duration of cough recordings ranged from 0.23 to 6.59 s and was standardised into equal-length segments for model training.

Table 2 shows significant differences in the average duration of cough sounds (≈1–2 s) and respiratory sounds (≈13–30 s). It is envisaged that a shorter cough sound may be focused on capturing the intensity and the quality of the cough, while a longer respiratory sound could provide a more comprehensive representation of the breathing patterns.

As for the resampling rate, the cough sound files have a high sampling rate of 48,000 Hz, which is generally typical of high-quality audio and has the capability to capture a wide range of frequencies. By contrast, the respiratory sounds are sampled at 4000 Hz, which is much lower and may therefore miss finer details in the recorded audio [12]. This could be representative of different recording equipment or purposes, with cough sounds potentially requiring higher fidelity for accurate analysis.

### 2.2. Data Analysis

In this study, we have adopted waveform processing as a fundamental analysis tool for audio files. In particular, waveform analysis provides details of the visual representation of the audio signal’s amplitude over the passage of time. It may also help in identifying the loudness, the duration of the sound segments, and the presence of silence and/or noise. This is particularly useful in understanding the dynamics of the audio, such as the variations in sound intensity and temporal patterns, which can represent specific disease features for different asthma patients [13].

To enhance interpretability, the waveform plots have been updated to present the *x*-axis in seconds instead of sample points. This change provides clearer insight into the temporal characteristics of the sound signals.

The dynamic range of Figure 2, with a maximum amplitude of +32,767 and minimum amplitude of 24,217, reflects the varying intensity of breathing sounds. The peaks in the waveform represent louder respiratory phases, likely corresponding to inhaling or exhaling. However, quieter segments could indicate softer breathing or pauses between breaths. Figure 3, Figure 4 and Figure 5 depict the waveform analysis of sample audio from the respective dataset. In summary, the audio recording exhibits a broad dynamic range and a low-frequency profile characteristic of respiratory sounds. The variation in amplitude may reflect different phases of breathing, while the low-frequency dominance is typical of such sounds. This type of analysis can be valuable in medical contexts, including monitoring respiratory health, diagnosing conditions, and assessing treatment efficacy [14].

### 2.3. Signal Transformation Approaches

To develop an automated, lightweight model for asthma detection with respiratory and cough sounds, the audio is passed through a time–frequency transformation process utilising a spectrogram as a visual way to represent the audio’s frequency information over the passage of time. In general, a spectrogram is a visual representation of the frequency spectrum of a sound signal, depicting three dimensions—time, frequency, and amplitude. It is expected that patients *with* and *without* asthma are likely to demonstrate different spectrum features within the prescribed respiratory and cough sounds and, therefore, a spectrum analysis can enable the model to determine its efficacy based on any of the two specific data inputs.

Figure 6 shows the 2D spectrogram of a cough sound using the Gabor transform. Note that the *x*-axis represents time (seconds), the *y*-axis represents frequency (Hz), and the colour intensity indicates the signal amplitude in decibels (dB).

It is noteworthy that this research utilises three primary techniques for interpreting these non-auditory signals: the STFT, Gabor, and the CWT. All of these approaches are designed to transform the auditory signals into two-dimensional images or characteristic representations that are compatible with their integration into various models [15].

Firstly, STFT is a mathematical technique used in signal processing to determine the sinusoidal frequency and the phase content of the local sections of a time-series signal. The primary feature of this signal comprises a series of Fourier transforms of a windowed signal with the window being shifted along the time axis for each calculation stage [16]. The STFT is defined in Equation (Equation 1) [17,18].(1)$$STFT\left\{x\left(t\right)\right\}\left(f,\tau \right)=\int_{-\infty }^{+\infty }x\left(t\right)\;w\left(t-\tau \right)\;e^{-j2\pi ft}\;dt$$
where x(t) is the input signal, w(t−τ) is the window function, centred at time τ. *f* is the frequency variable. $e^{-j2\pi ft}$ is the complex exponential function, representing a sinusoid. The integral is taken over time. In this equation, the window function w(t−τ) is critical as it localises the signal in time, allowing the Fourier transform to provide frequency information specific to the time segment around τ. The result is a function of both time (τ) and frequency (*f*), providing a time–frequency representation of the original signal x(t).

Secondly, CWT is a specific analysis method used in signal processing to extract the frequency information from the time series signal [19]. CWT aims to employ wavelets, that is, small waves that grow and decay within a confined duration. These wavelets are scaled and translated versions of a mother wavelet, offering a more flexible approach to analysing signals with varying frequency content over time. The CWT of a continuous signal x(t) is defined in Equation (Equation 2):(2)$$CWT\left(\tau ,s\right)=\frac{1}{\sqrt{\left|s\right|}}\int_{-\infty }^{\infty }x\left(t\right)\;\psi^{*}\left(\frac{t-\tau }{s}\right)\;dt$$
where x(t) is the input signal. ψ(t) is the mother wavelet. $\psi^{*}\left(t\right)$ is the complex conjugate of the mother wavelet. τ is the translation parameter. *s* is the scale parameter. The integral is taken over time. Note that the function $\psi \left(\frac{t-\tau }{s}\right)$ represents the wavelet function scaled by *s* and translated by τ.

Thirdly, **Gabor** analysis is a variant of STFT where the key difference lies in the use of a specific window function, the Gaussian window, which is a bell-shaped curve. This window is applied to the signal to isolate small sections for analysis. The Gaussian window is known for its minimal spread in both the time and frequency domains, allowing for a more precise time–frequency representation [4]. The Gabor transform of a signal x(t) is defined in Equation (Equation 3):(3)$$G\left(t,f\right)=\int_{-\infty }^{\infty }x\left(\tau \right)\;e^{-\pi {\left(\tau -t\right)}^{2}}\;e^{-j2\pi f\tau }\;d\tau$$
where x(t) is the input signal. $e^{-\pi {\left(\tau -t\right)}^{2}}$ represents the Gaussian window function, at time *t*. $e^{-j2\pi f\tau }$ is the complex exponential function, representing a sinusoid of frequency *f*. The integral is taken over time τ.

It is imperative to mention that this equation applies a Gaussian window to the signal and then computes the Fourier transform of the windowed signal. The result G(t,f) represents the time–frequency distribution of the signal, whereby we have the following:

*x-axis:* Represents time in seconds, indicating the progression of the signal over a 4-s duration.

*y-axis*: Denotes frequency in Hertz (Hz), ranging from 0 Hz to 25,000 Hz, which captures the frequency components of the signal.

*z-axis*: Indicates the amplitude or intensity of the signal at each time–frequency point, measured in decibels (dB).

Finally, the colour map provides a visual representation of the amplitude. In particular, the warmer colours (yellow) correspond to higher intensity levels and the cooler colours (purple) to lower levels, which could be representative of the distinct features of the disease under detection.

### 2.4. Feature Extraction

In this study, we have adopted lightweight models, namely, ShuffleNet, SqueezeNet, MobileNet and EfficientNet, to extract features from the spectrogram.

It is especially noted that these models were trained on the ImageNet platform, commonly used in several other studies [10,20,21,22]. In particular, ImageNet is a large-scale, diverse dataset that has over 14 million labelled images widely used as a benchmark for the training and evaluation of deep learning models in the computer vision area. Its extensive coverage and pre-trained models can enable an effective transfer learning process, making it a foundational resource for advancing research in real-world applications. These lightweight models were also selected based on their computational efficiency, which is beneficial for mobile and edge devices with limited processing power. The specific importance of each lightweight model used in designing an automated system for asthma detection using respiratory and cough sound signals is as follows:**ShuffleNet** is a type of neural network architecture that is ideal for devices with limited computational resources, like smartphones or other mobile devices. The data flow through channels in layers. In ShuffleNet, the data from these channels are mixed up or “shuffled”. This mixing helps ShuffleNet learn better from the data without needing more power or memory [10]. ShuffleNet organises its data channels into groups. Within these groups, it does convolutions (filtering data) that help it learn from images or other inputs. By using groups, it can do many operations at the same time, which saves time and energy.**EfficientNet** is an innovative neural network architecture that sets new benchmarks for both efficiency and ACC. EfficientNet uses compound scaling to uniformly scale the network’s depth, width, and resolution using a set of fixed scaling coefficients. This approach not only increases the depth and improves the resolution but also ensures that the foundations become wider, so everything stays in proportion and works well together [23].**SqueezeNet** is designed to be both small in size and fast in performance, while still maintaining a high level of accuracy in tasks, requiring less computational power and storage. SqueezeNet allows it to do more with fewer parameters. The architecture uses squeezed layers that reduce the number of parameters, followed by expanded layers that increase them again. The building blocks of SqueezeNet are called fire modules, which are made up of squeezed and expanded layers. These modules are designed to keep the network efficient [24].**MobileNet** is designed with a focus on mobile and embedded vision applications, providing high-performance model architectures that can run efficiently on smartphones and other devices with limited computational resources. The core ideas in MobileNet involve the use of separable convolutions [25]. It breaks down the usual complex operations into simpler, smaller operations that are easier to compute. This results in a dramatic reduction in the number of computations and the model size. The architecture is modular, which means it is made up of building blocks that can be mixed and matched to create networks that suit different needs and capacities.

It is expected that each CNN model will extract 500 features concatenated with other models, so the features may be up to 2000 in all of the net combinations. And it processes its respective time–frequency representation to extract a set of features that are particularly suited to the characteristics captured by the transformation [26], resulting in a rich, diverse feature set for subsequent analysis. There are four CNN models, resulting in a total of fifteen combinations for feature extraction. Two kinds of feature selectors are provided, so there are 450 model trials shown in Table 3 in each dataset.

### 2.5. Feature Selection

This study has adopted the Relief method as an algorithm designed to assess the quality of the features in a dataset based on how well their values distinguish between instances that are near each other. Essentially, it works by repeatedly sampling an instance from the dataset and then finding its nearest neighbour from the same class (nearest hit) and from a different class (nearest miss) [27]. The relevance of each feature is updated based on how well it distinguishes the sampled instance from its nearest hit and nearest miss.

This study has also adopted the NCA method as a sophisticated feature selector in ML that aims to enhance classification performance by learning a distance metric to organise data more effectively. It prioritises crucial features by assigning weights, which helps in clustering similar instances closer together and distancing dissimilar ones. NCA focuses on maximising a function that quantifies the probability of correct classification based on nearest neighbours within the learned metric. The method involves using gradient descent to iteratively adjust feature weights to achieve optimal performance [28]. By emphasising important features and reducing dimensionality, NCA not only speeds up the training process but also improves the model’s ACC by mitigating the risk of overfitting. Although powerful, NCA requires significant computational resources and careful parameter tuning, making it ideal for applications with complex datasets where the relationships between features and outcomes are not straightforward.

In this case, 500 features are ranked in feature selection steps, and the original feature size is from 500 to 2000, depending on the feature extraction used. Feature selection helped to reduce computation in modelling and avoid over-fitting.

### 2.6. Classification Methods

The selected features were then fed into classical ML classifiers based on SVM, NN, RF, KNN, and DT algorithms. A summary and brief justification for the use of each classifier method are outlined in the following:**RF** is an ensemble learning method that operates by constructing multiple decision trees during training and outputting the class that is the mode of the classes of the individual trees [29].**SVM** is a classifier that finds the optimal boundary to separate different classes by maximising the margin between support vectors, which are the data points closest to the boundary [30].**NNs** are a set of algorithms, modelled loosely after the human brain, which are designed to recognise patterns and perform tasks like classification by learning from examples [31].**DT** is a flowchart-like tree structure where an internal node represents a feature(or attribute), the branch represents a decision rule, and each leaf node represents the outcome [32]. These classifiers are trained to detect the presence of respiratory diseases by recognising patterns and signatures in the cough and respiratory sounds that are characteristic of the condition.**KNN** is a simple, instance-based learning algorithm that classifies a sample based on the majority vote of its k nearest neighbours in the feature space [33].

Overall, the present study utilises five different classifier models that provide their own merits in developing an automated model for the detection of asthma.

### 2.7. Parameter Tuning

In this study, we use several ML models, including RF, NN, SVM, DT, and KNN. The choice of model parameters for each of these models is, therefore, paramount. These key model parameters were optimised through a grid search approach to achieve an optimal performance of ML models, and the range of model parameters is outlined below.

RF: Criterion: [‘gini’, ‘entropy’, ‘log_loss’]. N Estimators: [100, 200, 300] to stabilise predictions. Max Depth: [10, 20, 30, 40, 50]. Max Features: ‘sqrt’, to improve model diversity.SVM: C: [0.1, 1, 10, 100] to control regularisation strength. Kernel: [‘linear’, ‘poly’, ‘rbf’] for handling both linear and non-linear relationships.NN: Activation: [‘identity’, ‘logistic’, ‘tanh’, ‘relu’] to introduce non-linearity. Solver: [‘sgd’, ‘adam’] to explore different optimisation techniques. Hidden Layer Sizes: [(32), (64), (128), (32, 32), (64, 64), (128, 128)] for different network depths. Alpha: [0.0001, 0.001, 0.01, 0.1, 1.0] as a regularisation parameter to prevent over-fitting.DT: Criterion: [‘gini’, ‘entropy’, ‘log_loss’] as splitting criteria. Max Depth: [10, 20, 30, 40, 50] to control tree depth and prevent over-fitting. Min Samples Split: 10, to ensure each split has sufficient samples. Min Samples Leaf: 10, to avoid overly small leaves.KNN: N neighbours: [3, 5, …, 135] (odd numbers), to find the optimal number of neighbours.

### 2.8. Model Evaluation

Model performance for automated asthma detection was evaluated using 5-fold cross-validation and a set of key metrics, as follows: ACC, F1 score, precision, recall, and AUC. These metrics were also used for tuning model parameters. The mathematical definitions of these metrics are presented below.

**ACC**: This measures the proportion of true results (both true positives and true negatives) in the total number of cases examined [34]. It is defined in Equation (Equation 4).(4)$$\mathrm{Accuracy}=\frac{TP+TN}{TP+TN+FP+FN}$$
where we have the following:-TP = True Positives-TN = True Negatives-FP = False Positives-FN = False Negatives**Precision**: This is also known as PPV; quantifies the proportion of correctly predicted positive cases among all predicted positives [34]. Mathematically, it is defined as follows:(5)$$\mathrm{Precision}=\frac{TP}{TP+FP}$$**Recall**: This is also known as sensitivity or the true positive rate; it measures the proportion of actual positives correctly identified [7]. It is defined as follows:(6)$$\mathrm{Recall}=\frac{TP}{TP+FN}$$**F1 score**: This is the harmonic mean of precision and recall, offering a balance between them [7]. It is defined as follows:(7)$$F1=2\cdot \frac{\mathrm{Precision}\cdot \mathrm{Recall}}{\mathrm{Precision}+\mathrm{Recall}}$$

As a final evaluation method of the proposed asthma detection system, we employ the ROC curve [35] and the AUC metric [36]. The ROC curve plots the true positive rate against the false positive rate across various classification thresholds, providing insights into the model’s discrimination ability. The AUC quantifies the overall performance, with values closer to 1.0 indicating superior classification capability. Given the use of multiple classifiers (DT, SVM, NN, RF, KNN), AUC comparisons help us to determine the most effective model for distinguishing asthma-related respiratory and cough sounds from non-asthmatic cases.

## 3. Results

### 3.1. Proposed Model Results with Respiratory Sound

Figure 7 compares the average performance of the five proposed ML classifiers applied across these different feature extraction methods, where the values range from about 82% to 98%. It is, therefore, notable that several feature extractors seem to yield relatively high ACC across the different transformations applied. This suggests that all lightweight CNN net combinations are effective in detecting asthma. However, CWT is consistently among the highest-performing transformations across various feature extraction methods. The transformation of CWT with the NCA selector does not consistently improve the average performance of the classifiers. For some feature extraction methods, the performance with the NCA selector is slightly lower or comparable to the performance with the Relief selector. The increased performance variability seen with Gabor and Gabor-NCA transformations may be attributed to their inherent sensitivity to the specific characteristics of the input features.

The results of Figure 7 highlight several noteworthy patterns with respect to model performance across different transformation–selector combinations. KNN stands out with exceptionally high ACC, reaching 98.55%, particularly when paired with STFT-Relief. However, this uniformity in performance, achieved with a fixed parameter setup (n_neighbours = 3), raises concerns about over-fitting or over-dependence on specific feature extraction and selection methods. In contrast, SVM and RF models exhibit more consistent performance across combinations, with peak accuracies of 94.10% and 93.77%, respectively, under Gabor-NCA and STFT-NCA. These results suggest that these models may offer better robustness and adaptability to different feature engineering strategies.

The role of feature extraction and selection methods is also apparent. For example, Gabor-NCA appears to be a strong combination, yielding high ACC across multiple models, including SVM, RF, and DT. On the other hand, NN and DT exhibit more variability and generally lower performance, with their best results at 93.55% and 93.17%, respectively. These observations underscore the importance of selecting suitable feature engineering techniques and hyperparameters for optimal model performance. Further analysis should focus on validating the reliability of KNN’s exceptional performance, exploring the impact of feature engineering choices, and refining the configurations of models like SVM and RF to improve their adaptability and generalizability.

Table 4 reveals that—in a respiratory sound dataset—while KNN demonstrates a high ACC (e.g., 98.55% with Gabor-NCA), its other evaluation metrics, such as F1 score, precision, recall, and AUC, are significantly less balanced. For instance, despite the high ACC, the F1 score and recall for Gabor-NCA + KNN are only 66.46% and 57.30%, respectively, indicating that the model struggles with certain classes or edge cases. Moreover, models like CWT and CWT-Relief exhibit lower accuracies (e.g., 96.02% and 94.52%) without compensating improvements in other metrics, further highlighting KNN’s limitations in achieving balanced performance.

On the other hand, SVM demonstrates better overall stability and balance across all metrics (see Table 4 and Table 5). Further, the Gabor + NCA + SVM model achieves an F1 score of 92.49%, precision of 91.04%, recall of 91.85%, and AUC of 93.06%, indicating robust and reliable performance. Even with different parameter configurations (e.g., C = 1, kernel = linear or C = 100, kernel = poly), SVM maintains consistent and competitive results. Therefore, while KNN may deliver high ACC, its lack of balance across other critical metrics makes SVM a more suitable and reliable choice for this dataset. Furthermore, Table 5 presents the performance metrics with 95% confidence intervals for the respiratory sound dataset. It is evident that the SVM model registered the highest accuracy, recision, recall and F1 score compared to the random forest, KNN, decision tree and neural network models.

### 3.2. Proposed Model Results with Cough Sound

The cough sound dataset consists of 994 and 881 recordings; the recordings were collected from 511 asthmatic individuals (103 males and 408 females; mean age 55.23 ± 14.97 years, age range 10–82 years) and 815 non-asthmatic subjects (509 males and 306 females, primarily healthy university students without a history of asthma), respectively. This dataset was collected using a mobile phone’s microphone, specifically a basic mobile phone with a sampling frequency of 48 kHz. For asthmatic participants, the duration of the recordings ranged from a minimum of 0.5 s to a maximum of 6.59 s. In contrast, the durations of the recordings from non-asthmatic subjects ranged from 0.23 s to 5.42 s. This dataset is a valuable resource for studying the acoustic characteristics of cough sounds and their medical applications in distinguishing between asthmatic and non-asthmatic participants.

Based on aggregated ACC for each transformation method, the analysis revealed the best and the highest average ACC attained using the Gabor transformation to be 80.77% and 80.95%, respectively. When using the CWT transformation method, we obtained an average ACC of 74.39% and the highest ACC of 79.74%; using STFT resulted in an average ACC of 74.42% and the highest ACC of 74.84%.

Figure 8 compares the performance of the three transformations—Gabor, STFT, and CWT—on a cough sound dataset using the following four key metrics: accuracy, precision, recall, and F1 score. Among these, Gabor demonstrates the highest performance with an accuracy of 80.77%, precision has an accuracy of 81.20%, recall has an accuracy of 80.10%, and the F1 score has an accuracy of 80.60%, making it the most effective transformation for this dataset.

It is interesting to note that both STFT and CWT also deliver comparable results, with STFT achieving an accuracy of 74.42%, precision of 73.10%, recall of 74.00%, and an F1 score of 73.50%. Similarly, CWT records an accuracy of 74.39%, precision of 72.50%, recall of 75.10%, and an F1 score of 73.80%. While their overall performances are consistent, CWT’s higher recall suggests it identifies positive cases slightly better than STFT but at the expense of lower precision.

Overall, we show that the Gabor transformation method stands out as the most suitable transformation for this dataset, significantly outperforming STFT and CWT in all metrics. This suggests Gabor’s robustness in capturing meaningful features from cough sound signals. Further optimisation of STFT and CWT could potentially narrow the performance gap and enhance their suitability for similar tasks.

Table 6 highlights the performance of different transformation and feature selection combinations on the cough sound dataset, measured by average and highest ACC. The Gabor-Relief combination stands out with the highest average ACC (80.79%) and peak ACC (80.95%), closely followed by Gabor-NCA. In contrast, combinations like CWT-Relief and STFT-Relief show relatively lower performance. Overall, the Gabor transformation paired with either Relief or NCA consistently delivers the best results.

As presented in Figure 9, we now evaluate the ACC (%) values of five ML models (NN, KNN, RF, DT, and SVM) applied across six transformation-feature selection combinations (CWT-NCA, CWT-Relief, Gabor-NCA, Gabor-Relief, STFT-NCA, STFT-Relief). Among these, KNN achieves the highest ACC of 92.07% for the CWT-NCA combination, but its performance is inconsistent, dropping to 78.45% for STFT-Relief. NN and RF perform consistently, with Gabor-NCA yielding the best results for both (83.26% and 83.62%, respectively). DT, however, shows the weakest performance overall, with ACC peaking at only 76.87% for Gabor-NCA, making it the least suitable model.

In contrast, SVM demonstrates the most balanced and robust performance, achieving high ACC across all combinations, with its best result at 84.03% for Gabor-NCA and maintaining strong results for STFT-Relief (80.50%). Combinations involving the Gabor transformation (e.g., Gabor-NCA, Gabor-Relief) consistently yield the highest ACC for most models, indicating their suitability for this dataset. Based on this analysis, SVM paired with Gabor-NCA emerges as the optimal choice for achieving reliable and consistent results.

Table 7 reveals a significant disparity between the ACC and other metrics for KNN models. While the best-performing KNN configuration achieves an ACC of 92.07% with the CWT transformation, NCA selector, and feature extractor 34, its other metrics are much lower—F1 score (59.81%), precision (59.82%), and recall (62.85%). This pattern persists in other KNN configurations, such as one with 89.62% ACC, where the F1 score drops to 42.32%, precision to 54.51%, and recall to 38.73%, or configuration with 85.76% ACC, which records an F1 score of only 30.45%, precision of 51.98%, and recall of 26.92%. These results highlight a systemic issue with KNN, where high ACC often comes at the expense of other critical metrics, making it unsuitable for tasks requiring balanced predictions.

In contrast, SVM models paired with the Gabor transformation and NCA selector exhibit much more balanced performance across all metrics. For example, the SVM configuration with feature extractor 24 achieves an ACC of 83.31%, alongside a strong F1 score (80.07%), precision (75.93%), and recall (80.41%). Similarly, the SVM configuration with feature extractor 124 maintains an ACC of 84.03%, with an F1 score of 80.23%, precision of 76.85%, and recall of 79.13%. These balanced results demonstrate SVM’s ability to generalise effectively while avoiding the trade-offs seen in KNN. Given its consistent and reliable performance across all metrics, SVM with the Gabor transformation and NCA selector is the preferred choice for this dataset, making it better suited for real-world tasks requiring balanced and dependable predictions. This result has also been reinforced in Table 8 where the SVM model demonstrates the highest accuracy, precision, recall and F1 score.

## 4. Discussion

To the best of the authors’ knowledge, this is the first work to propose a unified model for asthma detection using both cough and respiratory sounds. Previous studies have focused on either cough-based or respiratory sound-based analysis, requiring separate models and potentially even pre-processing steps. In contrast, our approach eliminates the need to distinguish between sound types prior to analysis. This enables continuous input of audio data without the computational overhead of sound-type classification, making the method more efficient and cost-effective. Moreover, by removing this intermediate decision step, we reduce the potential for error propagation, which might enhance the system’s overall robustness and reliability.

For respiratory sound data, Xu et al. [37] achieved a significant performance improvement, reporting an ACC of 94.05%, compared to the performance reported by Haider et al. [38] and Nabi et al. [39]. For cough sound data, the model achieves an ACC of 83.31% on a significantly larger dataset (1943 samples), compared to work by Infante et al. [40] and Balamurali et al. [41], which reported higher AUC values of 94% and 91.2%, respectively, but on much smaller datasets. This highlights the robustness of the model in handling large-scale data, despite its slightly lower performance metrics compared to models using logistic regression or pDNN. Xi et al. [42] explored exhaled aerosol patterns for non-invasive lung disease diagnosis. While disease differentiation remains challenging, their approach—integrating fractal analysis with SVM—achieved 100% ACC on a small dataset and 99.1% on a larger one, demonstrating strong potential for precise, non-invasive asthma detection.

### 4.1. Key Findings

It is important to highlight that the results from both datasets—cough sound and respiratory sound—indicate some common trends. While KNN models exhibit high ACC in some configurations, their other metrics, such as F1 score, precision, and recall, remain unbalanced, making them less suitable for reliable and robust classification across both datasets. For example, in the respiratory sound dataset, KNN models show high ACC (e.g., 98.55% with Gabor-NCA), but their F1 scores and recall values are significantly lower, demonstrating poor performance on certain classes. Similarly, in the cough sound dataset, KNN models with various configurations also fail to provide balanced results, despite achieving high ACC.

Table 9 presents a comprehensive performance comparison between our proposed lightweight hybrid model and several state-of-the-art methods previously reported in the literature. Specifically, for respiratory sound data, our framework achieves an accuracy of 94.05%, which notably exceeds the reported AUC of approximately 83.6% from earlier studies employing SVM and ensemble classifiers [38,39]. This improvement in accuracy clearly highlights the efficacy of integrating lightweight CNN architectures with classical ML methods alongside optimised feature selection NCA, demonstrating superior diagnostic capability compared to existing techniques.

In cough sound-based asthma detection, our method achieved an accuracy of 83.31%. While this is slightly lower than the best-reported performances—91.2% accuracy by Balamurali et al. [41] using deep neural networks and 94.0% AUC by Infante et al. [40] using logistic regression—it is crucial to emphasise that our results were validated on a significantly larger dataset, comprising 1943 cough segments from 1326 individuals. This much larger and more heterogeneous dataset offers a realistic evaluation scenario, confirming the robustness and generalizability of our proposed model in practical clinical settings. Additionally, our lightweight approach substantially reduces computational complexity and resource requirements, making it especially suitable for deployment in resource-constrained or edge-computing environments.

In summary, our hybrid model not only surpasses existing respiratory sound analysis methods but also achieves competitive performance on cough-based diagnosis under significantly more demanding conditions. These findings validate the potential of our lightweight hybrid approach for practical and efficient asthma diagnostics.

### 4.2. Limitations

Despite these promising results, several limitations should be acknowledged. This study relied on a limited dataset, which may have affected the generalizability of the findings to a broader population. Additionally, while ensemble feature extraction and majority voting show effectiveness, this approach increases computational complexity and may not be feasible in real-time diagnostic settings. Moreover, the reliance on specific feature extractors such as SqueezeNet, MobileNet, and EfficientNet means that the results are influenced by the inherent strengths and weaknesses of these models, potentially limiting adaptability to other types of sound data.

### 4.3. Future Work

In future work, we will aim to explore ensemble methods, such as majority voting, to enhance the overall ACC and robustness of the classification model across both cough sound and respiratory sound datasets. By combining predictions from multiple models—each trained with different configurations of feature extractors, feature selectors, and classifiers—majority voting can help mitigate the weaknesses of individual models and improve decision consistency.

For instance, leveraging models with complementary strengths, such as KNN for high ACC and SVM for balanced performance across metrics, could provide a more reliable classification system. This approach has the potential to further improve the model’s generalizability and ensure that the system performs well under diverse real-world scenarios. Additionally, future research could investigate dynamic weighting in the voting mechanism to prioritise models with higher reliability for specific datasets or sound types.

The promising outcomes of this research show advancements in the early detection and management of asthma and COPD. Future research should focus on enhancing AI models to more effectively detect the early stages of these diseases [43]. This involves integrating larger and more diverse datasets that not only encompass a broader demographic but also include varied environmental exposures to improve the models’ robustness and diagnostic ACC across different populations.

To address the complexities of these respiratory conditions, it is crucial to expand the datasets to include early-stage asthma and COPD cases. Such datasets will enable AI models to learn the subtle variations and early markers of these diseases, which are critical for timely and accurate diagnosis [44].

Further development could explore hybrid models that combine different ML techniques, including advanced DL algorithms. These models would leverage the strengths of various approaches to improve the sensitivity and specificity of diagnostics, making them more effective in distinguishing between asthma, COPD, and other respiratory conditions.

Refining the majority voting system by implementing weighted voting strategies—where weights are assigned based on each model’s training performance—could enhance the overall diagnostic ACC. This approach would make the AI systems more reliable and robust, providing clinicians with tools that support better decision-making in clinical settings [45].

Moreover, the incorporation of real-time data processing capabilities could revolutionise diagnostic procedures by enabling instant and on-site disease detection. This is especially important for managing conditions like asthma and COPD, where early detection can significantly alter disease outcomes and improve patient management.

Additionally, embedding real-time data processing capabilities would revolutionise diagnostic procedures by facilitating immediate and on-site disease detection. This is particularly crucial for managing asthma, where the early detection of stage progression can significantly alter therapeutic approaches and improve patient outcomes [46].

## 5. Conclusions

This study marks a significant advancement in the AI-driven detection of respiratory diseases by combining cough and respiratory sound analysis. The proposed framework employs advanced machine learning techniques, including the SVM model with SqueezeNet and EfficientNet as a combined feature extractor (represented as “24”) and NCA for feature selection. This configuration delivers a well-balanced and reliable diagnostic performance across key evaluation metrics, making it an optimal solution for analysing both cough and respiratory sound datasets. The approach ensures robustness and adaptability, addressing the diverse characteristics of the two datasets while maintaining consistency in performance.

The integration of dual-sound analysis and majority voting ensemble techniques has proven highly effective, achieving precise and consistent diagnostic outcomes. Among the tested configurations, the SVM model with the SqueezeNet + EfficientNet (“24”) feature extractor emerges as the best choice for compatibility across datasets. It achieves a balanced performance in ACC, F1 score, precision, recall, and AUC, underscoring the model’s versatility and reliability in diagnosing conditions such as asthma and AUC.

Future work should focus on expanding datasets to include broader demographic and environmental variations and refining real-time diagnostic capabilities. The continued development of ensemble methods, such as majority voting with dynamic weighting, could further enhance the ACC and robustness of the system. These advancements have the potential to revolutionise respiratory healthcare practices, enabling earlier detection and better management of asthma, COPD, and other respiratory conditions on a global scale.

## Figures and Tables

**Figure 1 diagnostics-15-01155-f001:**
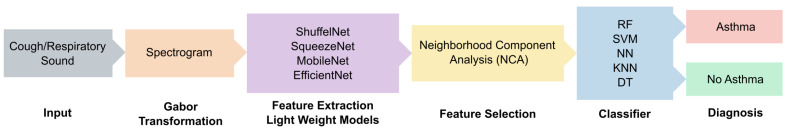
Block diagram of the proposed asthma detection system using respiratory and cough sound signals.

**Figure 2 diagnostics-15-01155-f002:**
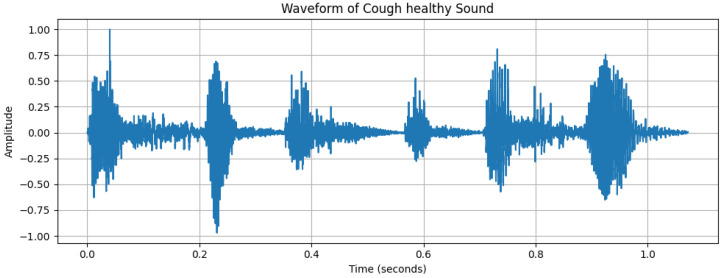
Waveform of a cough sound from an asthma subject, with the *x*-axis representing time (s) and the *y*-axis representing amplitude.

**Figure 3 diagnostics-15-01155-f003:**
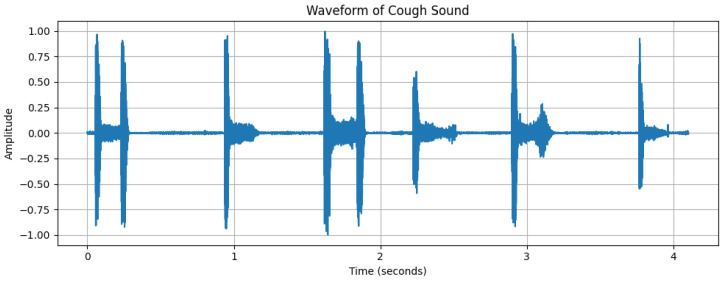
Waveform of a cough sound from a healthy subject, with the *x*-axis representing time (s) and the *y*-axis representing amplitude.

**Figure 4 diagnostics-15-01155-f004:**
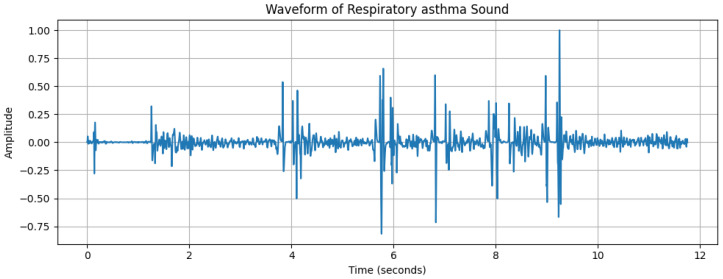
Waveform of a respiratory sound from an asthmatic subject, with the *x*-axis representing time (s) and the *y*-axis representing amplitude.

**Figure 5 diagnostics-15-01155-f005:**
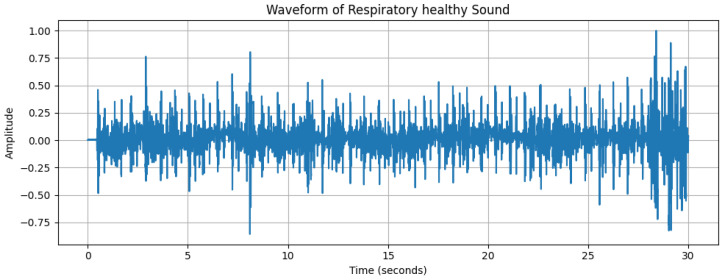
Waveform of a respiratory sound from a healthy subject, with the *x*-axis representing time (s) and the *y*-axis representing amplitude.

**Figure 6 diagnostics-15-01155-f006:**
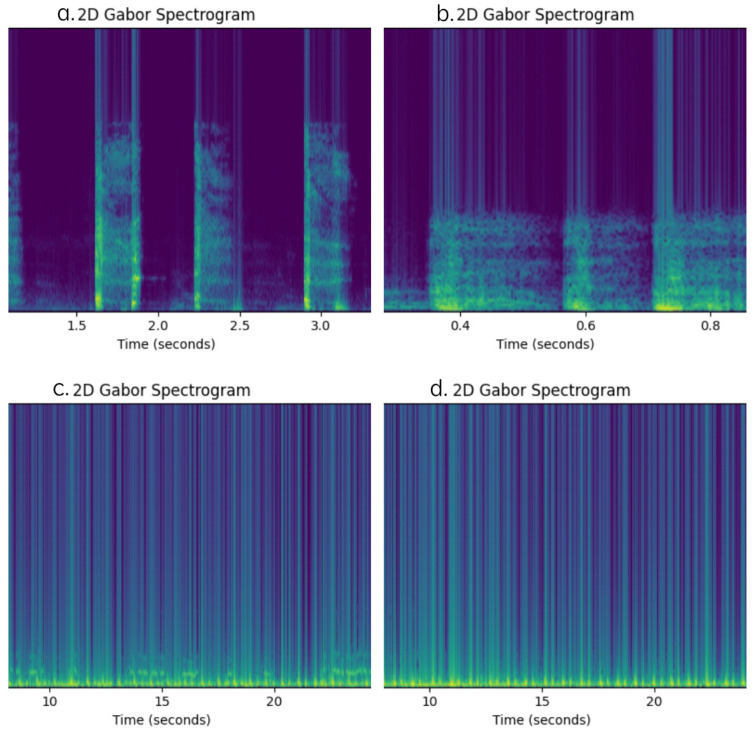
Gabor transform of the cough and respiratory signals, representing the spectra of ((**a**): **Top Left**) asthma cough audio, ((**b**): **Top Right**) healthy cough audio, ((**c**): **Bottom Left**), asthma respiratory audio, and ((**d**): **Bottom Right**) healthy respiratory audio.

**Figure 7 diagnostics-15-01155-f007:**
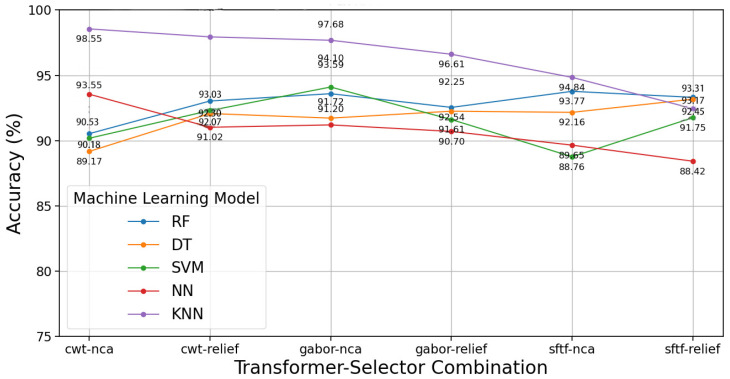
Average ACC obtained using classifiers and majority voting for the case of respiratory sound.

**Figure 8 diagnostics-15-01155-f008:**
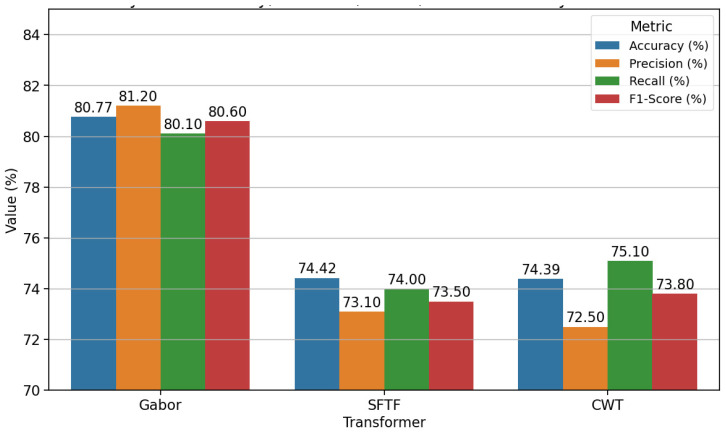
Model accuracy in terms of the transformation method and feature selection combinations for the model developed using cough sound signals.

**Figure 9 diagnostics-15-01155-f009:**
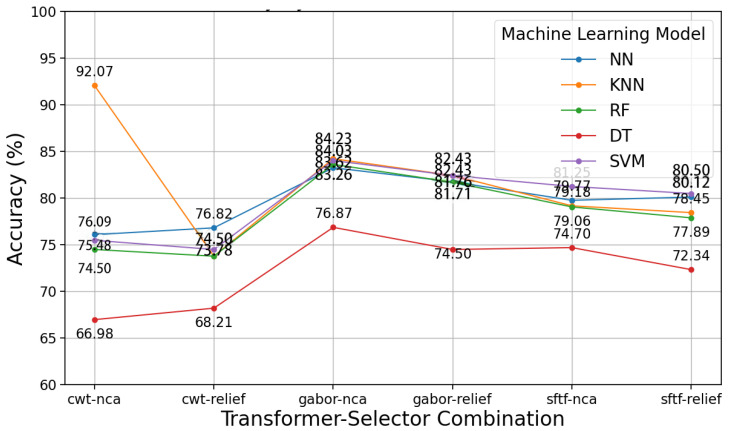
Model ACC in terms of the transformation method and feature selection combinations for the case of cough sound signals.

**Table 1 diagnostics-15-01155-t001:** Characteristics of cough and respiratory sound dataset used to develop an automated lightweight model for asthma detection.

Characteristics	Details
Duration	Duration of each audio file used to determine the range, the average, and the distribution of data length
File size	Assessing the file size to determine the dataset storage requirements
Sampling rate	The number of snapshots taken to recreate the original sound wave
Bit depth	The number of amplitude values in each snapshot to determine the audio resolution with the sampling rate

**Table 2 diagnostics-15-01155-t002:** Statistical analysis of the audio data used to build an automated lightweight model for asthma detection.

Metrics	Cough-Cough	Cough-Control	Resp-Asthma	Resp-Control
Avg duration (s)	1.77	1.60	17.07	29.93
Avg size (kB)	174	162	136	239
Sampling rate (Hz)	48,000	48,000	4000	4000
Bit depth	16	16	16	16

**Table 3 diagnostics-15-01155-t003:** The signal transformation method, feature extractor, feature selector, and the final classification model used to develop an automated lightweight model for asthma detection with respiratory and cough sounds.

Transformation	Feature Extractor	Feature Selector	Classifier
Gabor	ShuffleNet	Relief	SVM
	ShuffleNet+SqueezeNet		
	ShuffleNet+MobileNet		
	ShuffleNet+EfficientNet		RF
	ShuffleNet+SqueezeNet+ MobileNet		
CWT	ShuffleNet+SqueezeNet+EfficientNet		
	SqueezeNet“2”		DT
	SqueezeNet+MobileNet	NCA	
	SqueezeNet+EfficientNet		
	SqueezeNet+MobileNet+EfficientNet		NN
STFT	ShuffleNet+MobileNet+EfficientNet “134”		
	MobileNet		
	MobileNet+EfficientNet		KNN
	EfficientNet		
	All Nets		

**Table 4 diagnostics-15-01155-t004:** The performance metrics (%) of the proposed SVM and KNN models based on the respiratory sound signal. N = n_neighbour, EXT represents the number of features used, and SEL represents the feature selection method.

Transform	EXT	SEL	Model	ACC	Parameter	F1	Precision	Recall	AUC
Gabor	124	NCA	KNN	98.55	n_neighbour = 3	66.46	63.43	57.30	69.86
Gabor	1234	NCA	KNN	97.12	n_neighbour = 3	66.05	64.04	65.30	67.28
Gabor	12	NCA	KNN	97.10	n_neighbour = 3	58.45	53.94	70.56	60.09
CWT	12	Relief	KNN	96.02	n_neighbour = 3	61.93	57.13	72.86	61.66
CWT	13	Relief	KNN	94.52	n_neighbour = 3	59.13	58.41	64.17	63.13
Gabor	34	NCA	SVM	94.12	C = 1/kernel linear	82.11	83.36	77.73	85.54
Gabor	23	NCA	SVM	94.10	C = 100/kernel RBF	74.73	73.11	75.69	77.69
Gabor	24	NCA	SVM	94.05	C = 100/kernel poly	92.49	91.04	91.85	93.06

**Table 5 diagnostics-15-01155-t005:** The performance metrics with the 95% confidence interval (CI) for the respiratory sound dataset. The best model is boldfaced.

Model	Accuracy (95% CI)	Precision (95% CI)	Recall (95% CI)	F1 Score (95% CI)
**SVM**	94.05 ± 1.2%	93.8 ± 1.3%	93.6 ± 1.2%	93.7 ± 1.2%
Random Forest	92.70 ± 1.4%	92.5 ± 1.5%	92.3 ± 1.4%	92.4 ± 1.3%
KNN	91.80 ± 1.6%	91.5 ± 1.5%	91.2 ± 1.6%	91.4 ± 1.5%
Decision Tree	89.50 ± 1.9%	89.3 ± 1.8%	89.0 ± 1.9%	89.1 ± 1.8%
Neural Network	93.10 ± 1.3%	92.9 ± 1.3%	92.7 ± 1.2%	92.8 ± 1.3%

**Table 6 diagnostics-15-01155-t006:** The average and the highest accuracy obtained by the transformation method and feature selection.

Transformation-Feature Selection	Average ACC (%)	Highest ACC (%)
CWT-NCA	74.65	79.74
CWT-Relief	69.82	71.94
Gabor-NCA	80.79	80.95
Gabor-Relief	80.18	80.68
STFT-NCA	73.63	74.84
STFT-Relief	71.64	74.15

**Table 7 diagnostics-15-01155-t007:** The performance metrics (%) of the proposed SVM and KNN models based on cough sound signals. N = n_neighbour, EXT represents the feature extractor method, and SEL represents the feature selector method.

Transform	EXT	SEL	Model	ACC	Parameter	F1	Precision	Recall	AUC
CWT	NCA	134	KNN	85.76	n_neighbour = 3	30.45	51.98	26.92	54.86
CWT	NCA	23	KNN	88.60	n_neighbour = 3	43.09	54.98	40.34	57.26
CWT	NCA	123	KNN	89.62	n_neighbour = 3	42.32	54.51	38.73	57.76
CWT	NCA	124	KNN	90.12	n_neighbour = 3	48.81	58.81	45.04	63.87
CWT	NCA	24	KNN	90.12	n_neighbour = 3	32.69	54.12	30.24	56.99
CWT	NCA	34	KNN	91.12	n_neighbour = 3	48.45	56.82	47.06	60.83
STFT	Relief	34	KNN	92.07	n_neighbour = 3	59.81	59.82	62.85	65.77
Gabor	NCA	24	SVM	83.31	C = 10, kernel = rbf	80.07	75.93	80.41	75.90
Gabor	NCA	2	SVM	84.03	C = 10,				
kernel = rbf	79.84	76.66	78.57	76.63					
Gabor	NCA	124	SVM	84.03	C = 10,				
kernel = rbf	80.23	76.85	79.13	76.97					

**Table 8 diagnostics-15-01155-t008:** The performance metrics with the 95% confidence interval (CI) for the cough sound dataset. The best model is boldfaced.

Model	Accuracy (95% CI)	Precision (95% CI)	Recall (95% CI)	F1 Score (95% CI)
**SVM**	83.31 ± 1.5%	83.0 ± 1.6%	82.8 ± 1.5%	82.9 ± 1.4%
Random Forest	82.10 ± 1.6%	81.8 ± 1.5%	81.5 ± 1.6%	81.6 ± 1.5%
KNN	81.05 ± 1.8%	80.7 ± 1.7%	80.5 ± 1.8%	80.6 ± 1.7%
Decision Tree	79.50 ± 2.0%	79.1 ± 1.9%	78.8 ± 2.0%	78.9 ± 1.9%
Neural Network	82.75 ± 1.4%	82.4 ± 1.5%	82.2 ± 1.4%	82.3 ± 1.4%

**Table 9 diagnostics-15-01155-t009:** Comparison of the proposed framework with existing methods on cough and respiratory sound analysis.

Author,	Subtype	Classifier		Performance	Data	Data Size
Year		Type	Algo.			
Balamurali et al. 2019 [41]	Cough sound	DL	DNN	ACC = 91.2%	Private	51 subjects
Infante et al. 2017 [40]	Cough sound	ML	LR	AUC = 94	Private	54 patients
Xu et al. 2024 [37]	Cough sound	ML	SVM	ACC = 83.31%	Private	1943 cough sounds
Haider et al. 2019 [38]	Respiratory sounds	ML	SVM	AUC = 83.6	Private	30 COPD; 25 healthy subjects
Nabi et al. 2019 [39]	Respiratory sounds	ML	ensemble	AUC = 83.6	Private	55 asthmatic patients
Xu et al. 2024 [37]	Respiratory sounds	ML	SVM	ACC = 94.05%	Private	205 respiratory sounds
**Proposed work**	Cough sounds	Hybrid model for Normal vs. asthma	Feature extractor: Gabor-lightweight model Classifier: SVM	ACC = 83.31%	Private	1326 participants (511 asthma, 815 healthy) 1943 cough segments (511 asthma, 815 healthy)
	Respiratory sounds			ACC = 94.05%		

## Data Availability

The original contributions presented in the study are included in the article, further inquiries can be directed to the corresponding author.

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
