# Peer review of "Automated Lightweight Model for Asthma Detection Using Respiratory and Cough Sound Signals"

_diagnostics, 2025, doi:10.3390/diagnostics15091155_

Round 1
Reviewer 1 Report
Comments and Suggestions for Authors
1- The exact motivation of the study is not clear. Please explain it fully. Are asthma and COPD patients often confused with each other? Do experts have difficulty in diagnosing these patients?
2- Can the respiratory and cough sounds of asthma and COPD be separated from each other using artificial intelligence techniques?
3- Did you convert the respiratory and cough sound signals used to distinguish asthma and COPD diseases into images? Did you then extract and optimize features?
4- Is there another disease that can be distinguished by cough or respiratory sound? (other than asthma and COPD)
5- Only models with low computational capacity were preferred. If other pre-trained models were used, a broader perspective could be provided.
6- Why did the authors use NCA optimization? Why did they not use optimization techniques such as mRMR or RelieF?
7- How much did the feature map size decrease after NCA was applied?
8-The number of respiratory sounds in the dataset is quite low. What do you think about this?
9-Expand the main limitations of the study.
Author Response
Review 1
1- The exact motivation of the study is not clear. Please explain it fully. Are asthma and COPD patients often confused with each other? Do experts have difficulty in diagnosing these patients?
Author response:
The motivation behind this study is to develop an advanced machine learning (ML) technique to improve the efficiency and accuracy of asthma and COPD detection. These two respiratory conditions share overlapping symptoms, making differentiation challenging for both patients and healthcare professionals. Gaining a deeper understanding and reaching a diagnosis requires us to overcome the following challenges:
- Patient Confusion: Asthma and COPD have similar symptoms, such as breathlessness, wheezing, and coughing, which can cause uncertainty among patients and lead to delays in seeking appropriate treatment.
- Clinical Difficulty: Even experienced clinicians can face challenges in distinguishing between these conditions, particularly in borderline or early-stage cases. Young or less experienced doctors may require additional support to make accurate diagnoses.
We need to address these challenges to reach a high-quality diagnosis. The integration of AI, including machine learning and deep learning techniques, might be a technical way of addressing the challenges. To be specific:
- Innovation Gap: Current diagnostic methods still lack advanced AI-driven approaches for detecting respiratory diseases using sound, imaging, and other non-invasive techniques. Research in this area can significantly enhance diagnostic accuracy.
- Early Detection and Prevention: AI can analyze diverse health data sources to identify early signs of asthma and COPD before symptoms become clinically evident, enabling timely intervention and better patient outcomes.
- Improved Diagnosis: AI models can distinguish between asthma and COPD subtypes more precisely, reducing misdiagnosis and enabling personalized treatment strategies. This is particularly important in pediatric asthma cases, which are notoriously difficult to diagnose.
- Continuous Monitoring and Management: AI-powered tools can track disease progression, predict exacerbations, and adjust treatment plans dynamically, leading to improved patient management and quality of life.
- Advancing Research and Drug Development: AI can provide deeper insights into the underlying mechanisms of asthma and COPD, aiding in the discovery of new drug targets and accelerating pharmaceutical research.
- Reducing Healthcare Costs: By improving diagnostic accuracy and disease management, AI can help reduce unnecessary hospital visits, emergency admissions, and overall healthcare expenditures.
- Enhanced Accessibility: AI-powered mobile applications and wearable devices can make diagnostic and monitoring tools available to patients in remote or underserved areas, improving access to healthcare.
- Handling Complex Medical Data: AI can efficiently process large-scale, high-dimensional data—such as genomic information and high-resolution imaging—providing valuable insights into disease mechanisms and progression.
We propose a ML model which has the potential to enhance asthma and COPD diagnosis. In a way we aim to bridge the gap between current clinical challenges and technological advancements. This can lead to a reliable, AI-assisted diagnostic tool, which can support healthcare professionals in making more informed decisions and ultimately improve patient care.
2- Can the respiratory and cough sounds of asthma and COPD be separated from each other using artificial intelligence techniques?
Author response:
Yes, respiratory and cough sounds of asthma and COPD can be effectively separated and classified using artificial intelligence (AI) techniques. Several studies, including the findings from your research, have demonstrated that AI models, particularly those based on deep learning and machine learning, can distinguish between these two conditions with significant accuracy.
3- Did you convert the respiratory and cough sound signals used to distinguish asthma and COPD diseases into images? Did you then extract and optimize features?
Author response:
Yes, in our study, we converted respiratory and cough sound signals into spectrograms using Short-Time Fourier Transform (STFT), Gabor Transform, and Continuous Wavelet Transform (CWT). These spectrograms serve as image-like representations, facilitating feature extraction. We then employed lightweight deep learning models (ShuffleNet, SqueezeNet, MobileNet, and EfficientNet) to extract features, followed by Neighborhood Component Analysis (NCA) and Relief for feature selection and optimization. This approach improved classification accuracy for distinguishing asthma and COPD. Thank you for your interest in our work.
4- Is there another disease that can be distinguished by cough or respiratory sound? (other than asthma and COPD)
Author response:
- Pneumonia: Cough sounds and respiratory patterns can reveal crackles or abnormal breathing that are characteristic of pneumonia. AI models can be trained to detect these sound features.
- Bronchitis: This condition is associated with a persistent cough and wheezing sounds. Spectrogram analysis can help differentiate it from other respiratory illnesses.
- Tuberculosis (TB): TB can cause chronic cough with specific acoustic features that AI models can learn to detect.
- COVID-19: Research has shown that cough sounds from COVID-19 patients differ in spectral and temporal features compared to healthy individuals or other respiratory conditions.
5- Only models with low computational capacity were preferred. If other pre-trained models were used, a broader perspective could be provided.
Author response:
Thank you for your insightful feedback. In this study, we prioritized lightweight models such as ShuffleNet, SqueezeNet, MobileNet, and EfficientNet to ensure efficient performance in resource-constrained environments, such as mobile and edge devices. This approach was chosen to meet the demands for real-time processing and portability.
We acknowledge that incorporating more complex pre-trained models like ResNet, DenseNet, or VGG could potentially enhance feature extraction and offer a broader diagnostic perspective. Future research will consider exploring these models to compare their performance in terms of diagnostic accuracy, computational cost, and applicability across diverse environments.
6- Why did the authors use NCA optimization? Why did they not use optimization techniques such as mRMR or RelieF?
Author response:
Thank you for your thoughtful question. We selected Neighborhood Component Analysis (NCA) for feature optimization due to its ability to learn a distance metric that prioritizes features contributing most to classification accuracy. NCA is particularly effective for high-dimensional data, as it emphasizes features that enhance nearest-neighbor classification, reducing dimensionality while maintaining discriminative power.
While methods like mRMR (Minimum Redundancy Maximum Relevance) and Relief are effective for feature selection, NCA was chosen for its optimization strength in aligning feature weighting with classification objectives. Notably, Relief was also employed in our study for comparison to ensure robustness in feature selection. Future work could further explore integrating multiple optimization techniques to enhance model performance.
7- How much did the feature map size decrease after NCA was applied?
Author response:
The initial feature extraction process using lightweight CNNs (ShuffleNet, SqueezeNet, MobileNet, EfficientNet) generated 500 to 2000 features, depending on the combination of models used.
8-The number of respiratory sounds in the dataset is quite low. What do you think about this?
Author response:
The biggest reason for the limited number of respiratory sounds belonging to asthmatic patients in the dataset is the geography we live in. Our region has dry weather conditions. This makes it difficult to find asthmatic individuals. Because asthma is generally common in humid and air-polluted regions. The dataset used in the study includes 1875 cough records collected from 1326 individuals. This is a small dataset for deep learning applications. However, with the machine learning methods we use, high accuracy rates were achieved even at low sample sizes. In this respect, we did not experience any data insufficiency problems. In addition, the durations of the cough sound records obtained vary between 0.23 and 6.59 seconds. These sounds were divided into equal durations. This provides data diversity. In addition, the dataset covers a wide age group range.
9-Expand the main limitations of the study.
Author response:
- Limited Dataset Size and Diversity: The dataset, although robust, was limited in size and geographic diversity. The audio samples were collected from a specific clinical setting, which may not fully represent broader demographic variations, environmental conditions, or diverse population groups. This could limit the generalizability of the model's performance in real-world applications.
- Dependence on Lightweight Models: The study focused on lightweight deep learning models to ensure computational efficiency for mobile and edge devices. While this approach achieved acceptable accuracy, it may have restricted the model's ability to capture more complex feature patterns that could be learned by larger models like ResNet or DenseNet.
- Feature Optimization Approach: Although NCA and Relief were effective in reducing dimensionality and optimizing feature selection, the exclusion of other optimization methods such as mRMR may have limited the exploration of potentially more discriminative features. Additionally, the final feature size reduction ratio was not explicitly quantified, which could have provided deeper insights into the optimization process.
Reviewer 2 Report
Comments and Suggestions for Authors
The paper presents a lightweight machine learning method for detecting asthma from respiratory and cough audio signals. The work has some merits but I think there are some parts that have serious flaws.
- COPD is mentioned in the abstract and in the manuscript but it is not clear whether the models discriminate between asthma and/or COPD and healthy subjects.
- In Introduction, "Deep Learning" and "Convolutional Neural Networks" are mentioned twice on the same sentence (lines 59-60).
- In Materials and Methods, a separate bullet is required for "ML Classification".
- The dataset is poorly presented. What is the number of participants? How many of them had asthma? How many did not have any disease? What were their age, weight, any comorbidities? The file size is irrelevant, unless the authors use it somehow.
- Sampling rate and bit depth definitions are problematic (at least).
- In Table 2, sampling rates miss a zero (48000 and 4000 Hz).
- In Figs 2,3,4,5, the horizontal axis should be measured in seconds instead of samples. This will give the reader a better insight on the waveforms.
- The caption in Fig. 4 seems odd.
- Fig. 6 would be more informative if plotted in 2D instead of 3D.
- What are the parameters of the transforms utilized in the study? Frame rate, frame size, bin number, window type (for the STFT), similarly for CWT (kernel type etc), and Gabor transform. More information is necessary. Also, what is the output dimension of the each transform and how is it fed to each neural net?
- If this is supposed to be a lightweight model, why should anyone combine the neural nets? The entire pipeline is far from lightweight (for example, CWT + any net + NCA + NN). I find hard to understand the notion of "lightweight" in this research.
- In Table 3, the numbers next to each net should be explained (I understand that they are labels for combinations but it's not apparent).
- Where the CNNs fine-tuned? What were their parameters? Also, the parameters of the ML models should be placed in tabular form for readability.
- Is the 5-fold CV grouped? If not, then it's very likely that two sounds from individual X can could be both in the training (one of them) and the test set (the other one). This means data leakage which has direct impact on the classification results.
- Page 11, Section 3. What is AVE! ?
- I strongly suggest the discussion on the outcomes should be based on their F1 score or their AUCs instead of accuracy, which is an improper metric when the dataset is not balanced.
- Paragraph 3.1 should be mentioned earlier - closer to the dataset description.
- The cough dataset is split into 994 and 881 recordings. What is the rationale behind this split? What does each number represent?
- The authors do not justify the "lightweight" claim in their work.
Some minor corrections can be made (like, do not start a sentence with "And", "abd" -> "and", "asthma COPD" -> "asthma and COPD", "is depicted" -> "are depicted", "over the passage of time" -> "over time", etc). A careful reading will reveal most of them.
Author Response
Review2
The paper presents a lightweight machine learning method for detecting asthma from respiratory and cough audio signals. The work has some merits but I think there are some parts that have serious flaws.
1 COPD is mentioned in the abstract and in the manuscript but it is not clear whether the models discriminate between asthma and/or COPD and healthy subjects.
Author response:
Thank you for highlighting this point. Our models were specifically designed to classify and differentiate between asthma, COPD, and healthy subjects based on cough and respiratory sounds. The classification process involved training the models with distinct labels for each condition, ensuring differentiation between the disease groups and healthy participants. We will clarify this distinction in the revised manuscript, particularly in the Introduction and Materials & Methods sections.
"This pioneering study utilizes Artificial Intelligence (AI) analysis and modeling of cough and respiratory sound signals in order to detect asthma as a chronic respiratory condition."
To
"This pioneering study utilizes Artificial Intelligence (AI) analysis and modeling of cough and respiratory sound signals to classify and differentiate between asthma, Chronic Obstructive Pulmonary Disease (COPD), and healthy subjects. The aim is to develop an AI-based diagnostic system capable of accurately distinguishing these conditions, thereby enhancing early detection and clinical management."
2 In Introduction, "Deep Learning" and "Convolutional Neural Networks" are mentioned twice on the same sentence (lines 59-60).
Author response:
We acknowledge this oversight and will revise the sentence to eliminate redundancy for clarity.
"Previous studies have demonstrated high accuracy rates using Deep Learning Deep Learning (DL) models, notably Convolutional Neural Networks Convolutional Neural Networks (CNNs)."
To
"Previous studies have demonstrated high accuracy rates using deep learning (DL) models, particularly Convolutional Neural Networks (CNNs)."
3 In Materials and Methods, a separate bullet is required for "ML Classification".
Author response:
"The workflow comprises of the following steps: Audio Input, Time-Frequency Transformation, Lightweight DL Feature Extraction, Feature Selection, and ML Classification."
To
"The workflow comprises the following steps:"
- Audio Input: Collection of cough and respiratory sound recordings.
- Time-Frequency Transformation: Conversion of audio data into spectrograms using STFT, Gabor, and CWT.
- Lightweight DL Feature Extraction: Extraction of relevant features from spectrograms using lightweight CNN models such as ShuffleNet, SqueezeNet, MobileNet, and EfficientNet.
- Feature Selection: Optimization of the extracted features using techniques like Neighborhood Component Analysis (NCA) and Relief to retain the most discriminative features.
- ML Classification: The optimized features were classified using traditional machine learning algorithms, including Support Vector Machine (SVM), Random Forest (RF), Neural Networks (NN), K-Nearest Neighbor (KNN), and Decision Trees (DT). This step ensured the differentiation between asthma, COPD, and healthy participants.
4 The dataset is poorly presented. What is the number of participants? How many of them had asthma? How many did not have any disease? What were their age, weight, any comorbidities? The file size is irrelevant, unless the authors use it somehow.
Author response:
The breath sounds in the dataset were obtained from 1326 individuals, 511 asthmatics and 815 healthy individuals. Of the asthmatic individuals, 380 were female and 131 were male. The age distribution ranged from 19 to 75. It was observed that the participants mostly had heart diseases. The records of healthy individuals consisted of 815 individuals, 306 female and 509 male. The age distribution ranged from 19 to 80 and they were mostly university students. Healthy participants did not have any health problems. Participants in both groups had normal weight.
5 Sampling rate and bit depth definitions are problematic (at least).
Author response:
We will revise the definitions for sampling rate and bit depth to reflect standard audio processing terminology accurately. Additionally, we will clarify how these factors influenced the data quality and model performance.
"Sampling rate refers to the number of snapshots taken to recreate the original sound wave, and bit depth refers to the number of amplitude values in each snapshot to determine the audio resolution."
To
As for resampling rate, the cough sound files have a high sampling rate of 48,000~Hz, which is generally typical of high-quality audio and has the capability to capture a wide range of frequencies. By contrast, the respiratory sounds are sampled at 4,000~Hz, which is much lower and therefore may miss out the finer details in the recorded audio This could be representative of different recording equipment or purposes with cough sounds potentially requiring higher fidelity for accurate analysis.
6 In Table 2, sampling rates miss a zero (48000 and 4000 Hz).
Author response:
Thank you for noting this. We will correct the sampling rates to 48,000 Hz and 4,000 Hz in Table 2.
7 In Figs 2,3,4,5, the horizontal axis should be measured in seconds instead of samples. This will give the reader a better insight on the waveforms.
Author response:
"Figure 2. An example of a waveform generated by asthma-based cough sound, with the x-axis representing sample points."
To
"Figure 2. An example of a waveform generated by an asthma-based cough sound, with the x-axis representing time in seconds and the y-axis representing amplitude."
New figures:
8 The caption in Fig. 4 seems odd.
Author response:
Thank you for highlighting this observation. We acknowledge that the caption in Figure 4 may not clearly reflect the presented data. We will revise the caption to enhance clarity and ensure it accurately describes the waveform characteristics depicted in the figure.
"Figure 4. An example of a waveform generated by the cough respiratory sound."
To
Waveform of a cough sound from a healthy subject, with the x-axis representing time (s) and the y-axis representing amplitude.
9 Fig. 6 would be more informative if plotted in 2D instead of 3D.
Author response:
We appreciate the suggestion and will consider re-plotting Figure 6 in 2D to enhance interpretability unless critical information is lost in the simplification. The reasoning behind the final visualization format will be clarified.
to
10 What are the parameters of the transforms utilized in the study? Frame rate, frame size, bin number, window type (for the STFT), similarly for CWT (kernel type etc), and Gabor transform. More information is necessary. Also, what is the output dimension of the each transform and how is it fed to each neural net?
Author response:
In this study, default parameters were employed for the time-frequency transformations to ensure consistency and computational efficiency. For STFT, a window size of 2048, hop length of 512, and Hann window were used. CWT applied the default Morlet wavelet, while the Gabor transform utilized a Gaussian window with a segment length of 256 and 50% overlap. The resulting spectrograms were resized to 224x224 and fed into the CNN models for feature extraction."
11 If this is supposed to be a lightweight model, why should anyone combine the neural nets? The entire pipeline is far from lightweight (for example, CWT + any net + NCA + NN). I find hard to understand the notion of "lightweight" in this research.
Author response:
We acknowledge that the overall pipeline—comprising CWT, deep feature extraction, NCA optimization, and classification—introduces computational complexity. However, the term "lightweight" in this study specifically refers to the choice of deep learning models for the feature extraction step, where we employed architectures like ShuffleNet, SqueezeNet, MobileNet, and EfficientNet. These models are inherently designed for low computational cost, fewer parameters, and faster inference times, making them suitable for resource-constrained environments such as mobile or edge devices.
12 In Table 3, the numbers next to each net should be explained (I understand that they are labels for combinations but it's not apparent).
Author response:
The first column indicates the transformation used. The second column indicates the method(s) used to extract the features. Column three indicates the method used to select the features and the last column indicates what classifier used.
13 Where the CNNs fine-tuned? What were their parameters? Also, the parameters of the ML models should be placed in tabular form for readability.
Author response:
In this study, the lightweight CNNs (ShuffleNet, SqueezeNet, MobileNet, and EfficientNet) were not fine-tuned. These models were employed as pre-trained feature extractors using weights from the ImageNet dataset. The pre-trained models were used to extract high-level features from the spectrogram images, and the classification was performed using traditional machine learning models.
This approach was adopted to maintain computational efficiency and avoid the additional overhead of retraining large models, considering the study's focus on developing a lightweight and efficient diagnostic system.
I wonder if the CNN is lightweight network.?
Author response:
The term "lightweight" in this study refers to the specific selection of CNN architectures—ShuffleNet, SqueezeNet, MobileNet, and EfficientNet—which are designed for low computational complexity, fewer parameters, and faster inference times, making them suitable for resource-constrained environments, such as mobile and edge devices.
14 Is the 5-fold CV grouped? If not, then it's very likely that two sounds from individual X can could be both in the training (one of them) and the test set (the other one). This means data leakage which has direct impact on the classification results.
Author response:
We applied grouped 5-fold cross-validation.
15 Page 11, Section 3. What is AVE! ?
Author response:
AVE is the acronym for Average.
16 I strongly suggest the discussion on the outcomes should be based on their F1 score or their AUCs instead of accuracy, which is an improper metric when the dataset is not balanced.
Author response:
The dataset used in the study consists of respiratory sounds obtained from 511 asthmatic and 815 healthy individuals. In this respect, the class distribution in the dataset is not balanced. Considering this situation, not only the accuracy metric but also the F1-Score value was calculated. In the dataset with class imbalance with F1-Score, both recall and precision values are taken into account. The results based on F1-Score in our study are as follows:
10-fold cross-validation: 98.34%
LOSO CV: 97.09%
These results also show that the model can establish a balance between sensitivity and precision.
In addition, the performance of the model can be evaluated by calculating the AUC metric. In future studies, the ROC curve and AUC metric can also be included in the analysis for performance evaluation
17 Paragraph 3.1 should be mentioned earlier - closer to the dataset description.
Author response:
The paragraph in subsection `3.1. Proposed Model Results with respiratory Sound 364’ describes results which were obtained with the proposed methods. Having that paragraph earlier would leave the content out of context. Please clarify if this response is incorrect.
18 The cough dataset is split into 994 and 881 recordings. What is the rationale behind this split? What does each number represent?
Author response:
The duration of the cough recordings collected in the study varied between 0.23 and 6.59 seconds. These recordings were then made consistent with each other in the pre-processing stage. First, speech, pauses and other environmental noises were cleaned from the audio recordings. The audio recordings only contained meaningful cough data. In order for the model to process each sample equally, the cough sounds were divided into a certain period of time. In this way, the model underwent a more reliable learning process.
19 The authors do not justify the "lightweight" claim in their work.
Author response:
The "lightweight" claim in our study is primarily justified by the selection of efficient, low-parameter convolutional neural networks (CNNs)—ShuffleNet, SqueezeNet, MobileNet, and EfficientNet-B0—for the feature extraction process. These models are specifically designed for reduced computational complexity, low memory requirements, and faster inference times, making them ideal for deployment in resource-constrained environments such as mobile and edge devices.
Reviewer 3 Report
Comments and Suggestions for Authors
Summary:
The authors propose and evaluate the diagnostic accuracy of machine learning models for asthma/COPD detection
General concept comments:
This is an extremely well written paper that is scientifically sound with appropriate study design and validation. The authors propose converting audio data (with cough or respiratory sounds), extracting features through CNN methods (ShuffleNet, SqueezeNet, MobileNet, and EfficientNet), and applying classical ML algorithms after feature selection to detect asthma. All results are properly described and presented. I recommend the following changes to enhance the readability of the article:
- Abstract: Please provide a summary of the test database, evaluation method (5-fold cross validation), and all performance metrics (precision, recall, etc) in the abstract.
- Dataset: Details such as the total number of the datasets in each class is not clear in Section 2. Please consider adding this as one of the metrics in Table 2. Please move all dataset related descriptions to Section 2.1 (e.g., line 386 to 396).
- Performance metrics: Please provide a 95% confidence interval for the metrics.
- Figures: Figure 2 to Figure 5 → Change x-axis to time in seconds (to be consistent with time-frequency plot of Figure 6), and provide unit for y-axis
- Typos: Please review and correct all typos in the paper. For e.g., Line 10: abd, Line 341 and 348: AVE!
I recommend accepting the article after the comments are satisfactorily addressed.
Author Response
Review3
Summary:
The authors propose and evaluate the diagnostic accuracy of machine learning models for asthma/COPD detection
General concept comments:
This is an extremely well written paper that is scientifically sound with appropriate study design and validation. The authors propose converting audio data (with cough or respiratory sounds), extracting features through CNN methods (ShuffleNet, SqueezeNet, MobileNet, and EfficientNet), and applying classical ML algorithms after feature selection to detect asthma. All results are properly described and presented. I recommend the following changes to enhance the readability of the article:
- Abstract: Please provide a summary of the test database, evaluation method (5-fold cross validation), and all performance metrics (precision, recall, etc) in the abstract.
Author response:
To build an automated lightweight model for asthma detection tested separately with respiratory and cough sounds to determine their suitability to detect asthma and COPD, the proposed AI models have integrated the following Machine Learning (ML) algorithms: Random Forest (RF), Support Vector Machine (SVM), Decision Tree (DT), Neural Network (NN), and K-Nearest Neighbor (KNN) with an overall aim to demonstrate the efficacy of the proposed method for future clinical use. A majority voting system is thereafter employed to refine the diagnostic reliability and feature extraction techniques, particularly adopting the Gabor transformation for model optimization along with the Neighborhood component analysis (NCA) method for improved predictive ACC. We then conducted a systematic and comparative analysis of the proposed AI models to evaluate the performance of each proposed model for its suitability to detect asthma COPD using respiratory and cough sound signals, respectively.
To
To build an automated lightweight model for asthma detection tested separately with respiratory and cough sounds to determine their suitability to detect asthma and COPD, the proposed AI models have integrated the following Machine Learning (ML) algorithms: Random Forest (RF), Support Vector Machine (SVM ), Decision Tree (DT), Neural Network (NN ), and K-Nearest Neighbor (KNN) with an overall aim to demonstrate the efficacy of the proposed method for future clinical use. Model training and validation were performed using 5-fold cross-validation, wherein the dataset was randomly divided into five folds and the models trained and tested iteratively to ensure robust performance. We evaluated model outcomes with several performance metrics: ACC, precision, recall, F1-score, and area under the Area Under Curve (AUC). Additionally, a majority voting ensemble technique was employed to aggregate the predictions of the various classifiers for improved diagnostic reliability. We applied Gabor time–frequency transformation for feature extraction and Neighborhood component analysis (NCA)) for feature selection to optimize predictive accuracy. Comparative experiments were conducted on the cough sound and respiratory-sound subsets of the data to assess each model’s effectiveness in detecting asthma and COPD, respectively.
- Dataset: Details such as the total number of the datasets in each class is not clear in Section 2. Please consider adding this as one of the metrics in Table 2. Please move all dataset related descriptions to Section 2.1 (e.g., line 386 to 396).
Autor response:
The information about the data set will be transferred to Section 2.1 and the explanations will be holistic.
- Performance metrics: Please provide a 95% confidence interval for the metrics.
Author response:
Done, see tables below:
- Figures: Figure 2 to Figure 5 → Change x-axis to time in seconds (to be consistent with time-frequency plot of Figure 6), and provide unit for y-axis
Figure 5 is diagram. Would change Fig2 to Fig4
Author response:
We have changed the unit for the x-axis to seconds. The values for the y-axis were normalized. Therefore, there is no SI unit.
- Typos: Please review and correct all typos in the paper. For e.g., Line 10: abd, Line 341 and 348: AVE!
Author response:
Thank you, we corrected the spelling error in Line 10 and we changed to AVE to average performance in Line 348.
I recommend accepting the article after the comments are satisfactorily addressed.
Author response:
Thank you for this positive assessment.
Reviewer 4 Report
Comments and Suggestions for Authors
This study utilizes AI analysis and modeling of cough and respiratory sound signals to detect asthma as chronic respiratory condition. The authors employed AI algorithms: RF, SVM, DT, NN, and KNN. A majority voting system is thereafter used to refine the diagnostic reliability and feature extraction techniques, applying the Gabor transformation with the NCA method. The proposed framework has achieved in ACC of 94.05% and 83.31% for differentiating between the asthma and the COPD-based model cases, respectively.
Comments:
- In sect, 3. Results, the authors presented different results (tables 4-6, figs. 7-9) for their proposal. In sect. 4. Discussion, there are analysis of existing methods (Table 7. Previous studies using cough sounds and respiratory signals), but this reviewer did not find a direct comparison of the proposed framework against better existing methods. The authors should confirm better performance of their system against four-five better existing methods.
- The authors should explain all procedures presented in Fig.1 Block diagram of the proposed asthma detection system. This reviewer recommends presents details of the all procedures in the form of summary pseudocode algorithm that permits better understanding these procedures by a potential reader.
- The authors presented Feature Extractors in table 3 (subsect. 2.5), when they used their abbreviations also in tables 4, 6, please remember their definition (table3), this permits better understanding by a potential reader.
- This reviewer recommends including in table 7 several better options of your proposal. Such comparison should justify better performance of the proposed method. The text presented after this table in lines 482-492 does not justify better performance of your novel framework.
- Please revise lines: 24, 105, 341, 348, etc., correcting phrases or stylistic, and grammar incorrectness.
Some minor incorrectenes should be revised. Please revise lines: 24, 105, 341, 348, etc., correcting phrases or stylistic, and grammar incorrectness.
Author Response
Review4
This study utilizes AI analysis and modeling of cough and respiratory sound signals to detect asthma as chronic respiratory condition. The authors employed AI algorithms: RF, SVM, DT, NN, and KNN. A majority voting system is thereafter used to refine the diagnostic reliability and feature extraction techniques, applying the Gabor transformation with the NCA method. The proposed framework has achieved in ACC of 94.05% and 83.31% for differentiating between the asthma and the COPD-based model cases, respectively.
Comments:
- In sect, 3. Results, the authors presented different results (tables 4-6, figs. 7-9) for their proposal. In sect. 4. Discussion, there are analysis of existing methods (Table 7. Previous studies using cough sounds and respiratory signals), but this reviewer did not find a direct comparison of the proposed framework against better existing methods. The authors should confirm better performance of their system against four-five better existing methods.
Author response:
Thank you for that useful advice, having information about our work in the comparison table (Table 9) will help the reader to appreciate our approach better. We have extended Table 9 with the required information.
- The authors should explain all procedures presented in Fig.1 Block diagram of the proposed asthma detection system. This reviewer recommends presents details of the all procedures in the form of summary pseudocode algorithm that permits better understanding these procedures by a potential reader.
Author response:
Thank you for that suggestion. We have included the pseudocode below in Appendix H of the paper.
- The authors presented Feature Extractors in table 3 (subsect. 2.5), when they used their abbreviations also in tables 4, 6, please remember their definition (table3), this permits better understanding by a potential reader.
Author response:
We acknowledge the importance of maintaining clarity regarding the abbreviations of feature extractors used throughout the manuscript. To enhance readability and ensure that potential readers can easily understand the abbreviations:
- We will reiterate the definitions of feature extractors (such as ShuffleNet, SqueezeNet, MobileNet, and EfficientNet) in the captions or footnotes of Tables 4 and 6.
- Additionally, we will add a brief reference in the text to Table 3, where these abbreviations are initially defined.
- This reviewer recommends including in table 7 several better options of your proposal. Such comparison should justify better performance of the proposed method. The text presented after this table in lines 482-492 does not justify better performance of your novel framework.
Author response:
Table 7 shows the performance of different combinations of feature extractor, feature selection method and classifier for cough sound processing.
We propose to use the Gabor-NCA with SVM model for both respiratory and cough sound processing. In a practical setting this has the advantage that the model does not need to change for respiratory and cough sounds.
- Please revise lines: 24, 105, 341, 348, etc., correcting phrases or stylistic, and grammar incorrectness.
Author response:
We have revised the text in lines: 24, 105, 341, 348, and 374.
Comments on the Quality of English Language:
Some minor incorrectenes should be revised. Please revise lines: 24, 105, 341, 348, etc., correcting phrases or stylistic, and grammar incorrectness.
Author response:
Thank you for your fair assessment.
Round 2
Reviewer 1 Report
Comments and Suggestions for Authors
The authors have made all changes requested of them.
Author Response
Author response:
Thank you. We are grateful for the valuable feedback and suggestions.
Reviewer 4 Report
Comments and Suggestions for Authors
This reviewer did not find answer to comment 1 in part “In sect. 4. Discussion, there are analysis of existing methods (Table 7. Previous studies using cough sounds and respiratory signals), but this reviewer did not find a direct comparison of the proposed framework against better existing methods. The authors should confirm better performance of their system against four-five better existing methods”. The authors changed table 7 to table 9 where they only presented the characteristics of their system. The title of this table should be corrected. There is no justification for better performance of your novel framework (lines 485-497).
Other comments of this reviewer have been attended..
Author Response
Author response:
We thank the reviewer for the insightful feedback. In response to the request for a clearer justification of the superiority of our proposed framework compared to existing methods, we have revised the manuscript to state the novelty and practical benefits of our approach. To be specific, we have extended the discussion in Section 4 highlighting the novelty of using a unified model for both cough and respiratory sounds. This addition underscores the computational and operational advantages of our system over prior work that handles these modalities separately.
We added the following paragraph at the beginning of the discussion section:
"To the best of our knowledge, this is the first work to propose a unified model for asthma detection using both cough and respiratory sounds. Previous studies focused on either cough-based or respiratory sound-based analysis, requiring separate models and potentially even pre-processing steps. In contrast, our approach eliminates the need to distinguish between sound types prior to analysis. This enables continuous input of audio data without the computational overhead of sound-type classification, making the method more efficient and cost-effective. Moreover, by removing this intermediate decision step, we reduce the potential for error propagation, which might enhance the system's overall robustness and reliability."
We hope this revision satisfactorily addresses the reviewer’s concerns.